# Quasiparticle Self-Consistent *GW* Study of Simple Metals

**DOI:** 10.3390/nano12203660

**Published:** 2022-10-18

**Authors:** Christoph Friedrich, Stefan Blügel, Dmitrii Nabok

**Affiliations:** Peter Grünberg Institut and Institute for Advanced Simulation, Forschungszentrum Jülich and JARA, 52425 Jülich, Germany

**Keywords:** simple metals, excited states, electronic structure, spectral function, quasiparticles, density functional theory, many-body perturbation theory, *GW* approximation, ab initio, computational methods

## Abstract

The GW method is a standard method to calculate the electronic band structure from first principles. It has been applied to a large variety of semiconductors and insulators but less often to metallic systems, in particular, with respect to a self-consistent employment of the method. In this work, we take a look at all-electron quasiparticle self-consistent GW (QSGW) calculations for simple metals (alkali and alkaline earth metals) based on the full-potential linearized augmented-plane-wave approach and compare the results to single-shot (i.e., non-selfconsistent) $G_{0}W_{0}$ calculations, density-functional theory (DFT) calculations in the local-density approximation, and experimental measurements. We show that, while DFT overestimates the bandwidth of most of the materials, the GW quasiparticle renormalization corrects the bandwidths in the right direction, but a full self-consistent calculation is needed to consistently achieve good agreement with photoemission data. The results mainly confirm the common belief that simple metals can be regarded as nearly free electron gases with weak electronic correlation. The finding is particularly important in light of a recent debate in which this seemingly established view has been contested.

## 1. Introduction

Over the last decades, the GW approximation has developed into a standard method for the calculation of the electronic band structure. It relies on many-body perturbation theory. The GW approximation to the electronic self-energy contains the exact-exchange potential and a large part of electron correlation through the usage of a screened electron interaction potential.

After Hedin [1] laid the theoretical foundation of the GW method in 1965, it took more than twenty years until the first calculations for real solid-state systems were published, independently by Hybertsen and Louie [2] and by Godby, Schlüter, and Sham [3]. They showed that the calculated fundamental band gap of covalently bound semiconductors fell within 0.1 eV of the experimental values. Since then, the GW method has been established to yield significantly improved fundamental band gaps and band structures (albeit not always with the accuracy reported in the early works) for a wide variety of semiconductors and insulators compared to standard density-functional calculations.

Not long after the first calculations of semiconductors, the GW method was applied to metals. Northrup and coworkers [4] demonstrated that the occupied bandwidth of bulk sodium, known to be overestimated in Kohn–Sham DFT, shrinks upon a one-shot (i.e., non-selfconsistent) GW self-energy correction but not sufficiently to reach an agreement with experiments. The authors then included an exchange-correlation kernel into the screened interaction as well as energy-only self-consistency, thus going beyond the one-shot approach, which eventually brought the theoretical bandwidth in close proximity to the experimental value. The occupied bandwidth of bulk lithium, the other material studied, was predicted to be 2.75 eV, close to that of Na. At the time, however, there was no experimental value to compare it with. Later, the bandwidth of Li was measured [5] to be 3.0 eV, a bit larger than their prediction but smaller than both the LDA and the one-shot GW values.

Thus far, while not as extensively as for band-gap materials, the GW approximation has been applied to a large variety of metallic systems, ranging from bulk transition metals [6,7,8,9,10,11], rare earths [12,13] over halfmetals [14,15] to topological semimetals [16] and many more. However, despite its significance for fundamental solid-state theory, the bandwidth problem of simple metals has not been re-investigated in a systematic way until very recently [17].

The authors of Ref. [17] compared the performance of several electronic-structure methods in the description of simple sp metals. Among other methods, they considered the one-shot GW approximation and embedded dynamical mean-field theory (eDMFT). Their results confirm that one-shot GW improves on the local-density approximation (LDA) description of Kohn–Sham DFT. In particular, in all cases, the GW bandwidth turns out to be closer to the experiment than the corresponding LDA value, although a difference to the experimental value remains. Furthermore, the authors investigated the role of local correlations using eDMFT and showed that it yields bandwidth values in better agreement to experiment values. This led the authors to claim that the electrons in simple metals cannot be regarded as forming nearly free electron gases and that a proper treatment requires local correlations (e.g., within eDMFT) to be taken into account, implying that simple metals cannot be considered weakly correlated systems, quite contrary to common belief.

In the present paper, we address this issue from a different perspective. We apply the quasiparticle self-consistent GW (QSGW) method [18] to the bulk phases of simple metals while leaving the electronic screening on the level of the random-phase approximation (i.e., without local correlations). The QSGW method is a self-consistent variant of the GW method, which not only allows the quasiparticle energies to change from iteration to iteration but also the quasiparticle wavefunctions. In this sense, it goes beyond the energy-only self-consistency mentioned earlier, as it also enables the electron density and the Hartree potential to be optimized in the iterations. The QSGW method has been shown to improve the description of many materials with respect to one-shot $G_{0}W_{0}$ calculations [18,19]. It has also been applied to some of the materials considered here, namely Na [19,20] and K [20]. However, in all cases, the QSGW results turn out to be larger and, thus, further away from the experiment than the one-shot $G_{0}W_{0}$ values of the present work.

To shed light on this issue, we have carried out systematic QSGW calculations for Li, Be, Na, Mg, K, Ca, Rb, Sr, and Cs and found the resulting bandwidths in close agreement with the experimental values. These results imply that the common understanding of the simple metals as nearly free electron gases with weak correlation is fundamentally correct. The success of eDMFT should, therefore, be attributed rather to the self-consistency incorporated in eDMFT than to the inclusion of local correlations.

The paper is structured as follows. Section 2 recapitulates the GW approximation and the QSGW approach with an emphasis on application to metals. The computational details of the calculations are summarized in Section 3. Section 4 presents and discusses the results of our calculations. Finally, we give the main conclusions of the study in Section 5.

## 2. Methods

The GW approximation is based on many-body perturbation theory with the interacting single-particle Green function
(1)$$G\left(r,r^{\prime };\epsilon \right)=\sum_{kn}\frac{\psi_{kn}\left(r\right)\psi_{kn}^{*}\left(r^{\prime }\right)}{\epsilon -E_{kn}+i\eta \;\mathrm{sgn}\left(\epsilon -\epsilon_{F}\right)}$$
as the central quantity. Here, $\psi_{kn}\left(r\right)$ and $E_{kn}$ are the quasiparticle wavefunctions and energies, respectively, $\epsilon_{F}$ is the Fermi energy, and η is a positive infinitesimal. Atomic units are used except where noted otherwise. According to Equation (Equation 1), the single-particle excitation spectrum is encoded in the pole structure of the Green function. As an auxiliary function, we introduce the non-interacting Green function, usually denoted by $G_{0}\left(r,r^{\prime };\epsilon \right)$. It is defined by Equation (Equation 1) with eigensolutions $\varphi_{kn}\left(r\right)$ and $\epsilon_{kn}$ of a mean-field system instead of $\psi_{kn}\left(r\right)$ and $E_{kn}$. The interacting Green function (Equation (Equation 1)) is related to the non-interacting Green function via the Dyson equation, which can be formulated as an effective quasiparticle equation of motion
(2)$$\left[-\frac{\nabla }{2}\right.+\left.V_{\mathrm{ext}}\left(r\right)+\int \frac{n(r^{\prime })}{|r-r^{\prime }|}d^{3}r^{\prime }\right]\psi_{kn}\left(r\right)+\int d^{3}r^{\prime }\;\Sigma \left(r,r^{\prime };E_{kn}\right)\psi_{kn}\left(r^{\prime }\right)=E_{kn}\psi_{k}\left(r\right)$$
with the external potential $V_{\mathrm{ext}}\left(r\right)$ and the electron number density n(r). The electronic self-energy is approximated by
(3)$$\Sigma \left(r,r^{\prime };\epsilon \right)=\frac{i}{2\pi }\int_{-\infty }^{\infty }d\epsilon^{\prime }\;G\left(r,r^{\prime };\epsilon +\epsilon^{\prime }\right)W\left(r^{\prime },r;\epsilon^{\prime }\right)e^{i\eta \epsilon^{\prime }}\;,$$
the so-called GW approximation, with the screened Coulomb interaction potential
(4)$$W\left(r,r^{\prime };\epsilon \right)=\;v\left(r,r^{\prime }\right)+\int v\left(r,r^{\prime \prime }\right)P\left(r^{\prime \prime },r^{\prime \prime \prime };\epsilon \right)W\left(r^{\prime \prime \prime },r^{\prime };\epsilon \right)\;d^{3}r^{\prime \prime }\;d^{3}r^{\prime \prime \prime }$$
and the bare interaction potential $v\left(r,r^{\prime }\right)=1/\left|r-r^{\prime }\right|$. For the polarization function, we employ the random-phase approximation
(5)$$P\left(r,r^{\prime };\epsilon \right)=\frac{-1}{\pi }\int_{-\infty }^{\infty }d\epsilon^{\prime }\;G\left(r,r^{\prime };\epsilon -\epsilon^{\prime }\right)\;G\left(r^{\prime },r;\epsilon^{\prime }\right)\;,$$
where a factor of 2 has been included to take account of the electron spin. Since the materials under investigation are all non-magnetic, an explicit spin dependence is omitted.

The GW self-energy, Equation (Equation 3), constitutes the leading-order term of an expansion of the self-energy with respect to the screened Coulomb interaction potential [21]. Plugging the first term of Equation (Equation 4) into Equation (Equation 3) yields the exact non-local exchange potential of the Hartree–Fock method. The remainder of Equation (Equation 4), when inserted into Equation (Equation 3), gives rise to a non-local dynamical scattering potential, which describes a large part of the many-body correlation effects. It is also responsible for the emergence of lifetime effects and plasmon satellites.

Equation (Equation 2) looks very similar to the Kohn–Sham equation. The only difference is seen in the term describing electron exchange and correlation, which, in Equation (Equation 2), contains the non-local and energy-dependent self-energy instead of the local and energy-independent exchange-correlation potential $v_{\mathrm{xc}}\left(r\right)$ of the Kohn–Sham equation. Another important difference is that the energy eigenvalues $E_{kn}$, in contrast to the Kohn–Sham eigenvalues, have a clear physical meaning. They can be interpreted as real excitation energies of the interacting system. The eigenvalues are complex-valued. In addition to the excitation energies, the imaginary part contains information about the lifetime broadening.

A more general way of presenting the quasiparticle band structure together with the lifetime broadening (and potential satellite features) is by plotting the k-resolved spectral function
(6)A(k,ϵ)=1πsgn(ϵF−ϵ)ImTrϵI−H0(k)−Σ(k,ϵ)−1
with the trace operator Tr, the identity matrix *I*, the Hartree Hamiltonian $H_{0}\left(k\right)$ (i.e., the matrix representation of the term in the square brackets of Equation (Equation 2)), and the matrix representation of the self-energy. Note that peaks in the spectral function are expected at energies ϵ, for which the expression in the square brackets vanishes or becomes minimal. Recognizing that $H_{0}\left(k\right)+\Sigma \left(k,\epsilon \right)$ is nothing else than the matrix representation of the left-hand side of Equation (Equation 2) reveals that the spectral function shows quasiparticle peaks at the solutions of Equation (Equation 2).

The product of two Green functions (Equation (Equation 1)) gives rise to a double band summation in Equation (Equation 5), which can be interpreted as summing over all single-particle excitations $n\to n^{\prime }$, where *n* and $n^{\prime }$ run, respectively, over all occupied and unoccupied states. In semiconductors and insulators, each band is either completely filled or completely empty. In metallic systems, however, there are bands that are only partially filled. Then, in order to sum over the excitations accurately, it is important to employ an interpolation method for $E_{kn}$. We use the tetrahedron method [22] for this purpose. The k summations are then treated as continuous k integrations with interpolated energies $E_{kn}$ between the explicit k points. Therefore, formally, an infinite system with infinitely many k points (and, consequently, infinitely many unit cells) is treated.

The tetrahedron interpolation method enables a precise sampling of the Fermi surface, which is also important for the Drude term, a special contribution to Equation (Equation 5), which originates from infinitesimal excitations within the same band (intra-band) from immediately below to immediately above the Fermi surface. This contribution arises only in metallic systems and can be written as [9,23]
(7)$$P_{\mathrm{intra}}\left(k,\epsilon \right)=\frac{k^{2}\omega_{\mathrm{pl}}^{2}}{4\pi }\left(\frac{1}{\epsilon^{2}}-i\pi \left|\frac{d\delta (\epsilon )}{d\epsilon }\right|\right)$$
with the plasma frequency $\omega_{\mathrm{pl}}$, which can be calculated from an integration of the absolute square of the momentum expectation value over the Fermi surface. The imaginary part of Equation (Equation 7) is irrelevant in practice, and only the real part has to be considered.

Equations (Equation 1)–(Equation 5) pose a hen-and-egg problem. The quasiparticle energies $E_{kn}$ and wavefunctions $\psi_{kn}\left(r\right)$ obtained from Equation (Equation 2) are needed to calculate the self-energy and also the electron number density of Equation (Equation 2) in the first place. This calls for an iterative solution, where one starts from a simple mean-field solution, e.g., from Kohn–Sham DFT and cycles through the equations until self-consistency is achieved. This is realized in the QSGW method.

The similarity of Equation (Equation 2) to the Kohn–Sham equation motivates us to make use of an existing DFT code for such an implementation. For this purpose, it is necessary to transform the non-Hermitian, energy-dependent self-energy operator to a Hermitian, energy-independent operator, which then also turns the complex-valued quasiparticle energies $E_{kn}$ into real eigenvalues. This transformation should be defined in such a manner that the new Hermitian self-energy is as close as possible to the original non-Hermitian self-energy. Originally, Faleev and coworkers [18] suggested the following transformation
(8)$$\Sigma_{nn^{\prime }}^{H}\left(k\right)=\frac{1}{4}\left[\Sigma_{nn^{\prime }}\left(k,\epsilon_{kn}\right)+\Sigma_{n^{\prime }n}^{*}\left(k,\epsilon_{kn}\right)\right.\left.+\Sigma_{nn^{\prime }}\left(k,\epsilon_{kn^{\prime }}\right)+\Sigma_{n^{\prime }n}^{*}\left(k,\epsilon_{kn^{\prime }}\right)\right]$$
where the self-energy is represented in the single-particle eigenfunctions $\varphi_{kn}\left(r\right)$ of the previous iteration, and $\epsilon_{kn}$ are the corresponding eigenvalues. Obviously, this definition makes the new self-energy operator $\Sigma_{nn^{\prime }}^{H}\left(k\right)$ Hermitian and energy-independent.

The self-consistent procedure is started with a self-consistent Kohn–Sham DFT calculation, where the exchange-correlation potential $v_{\mathrm{xc}}\left(r\right)$ can be regarded as an approximate self-energy. This calculation yields the single-particle eigenstates $\varphi_{kn}$ and $\epsilon_{kn}$, which serve as a first approximation to $\psi_{kn}$ and $E_{km}$ and with which the self-energy Equation (Equation 3) is evaluated. Solving Equation (Equation 2) with the resulting self-energy yields the (one-shot) $G_{0}W_{0}$ quasiparticle energies $E_{kn}$. Most GW studies stop at this point and are content with the one-shot quasiparticle solutions. The QSGW method proceeds by constructing the *Hermitianized* self-energy $\Sigma^{H}$ according to Equation (Equation 8), which replaces Σ of Equation (Equation 2) for the first QSGW iteration.

To concretize the notion of $\Sigma^{H}$ to be as close as possible to the non-Hermitian self-energy Σ, one can demand that the diagonal element of the effective Hamiltonian operator of Equation (Equation 2) should approximate the quasiparticle energy $E_{kn}$. However, with Equation (Equation 8), the diagonal element Ekn≈ϵkn+Re[Σnn(k,ϵkn)]−〈φkn|vxc|φkn〉 exhibits the wrong energy argument in the self-energy. To see this, add and subtract $v_{\mathrm{xc}}\left(r\right)\psi_{kn}\left(r\right)$ on the left-hand side of Equation (Equation 2) and treat $\Sigma \left(E_{kn}\right)-v_{\mathrm{xc}}$ as a small perturbation. Perturbation theory then gives Ekn=ϵkn+Re[Σnn(k,Ekn)]−〈φkn|vxc|φkn〉 for the real part, which differs from the previous expression by the energy argument of the self-energy. It was shown [16] that this inconsistency can cause an instability of the self-consistent cycle, potentially resulting in convergence towards an unphysical solution. Therefore, we employ the original definition Equation (Equation 8) only for the off-diagonal elements and use
(9)$$\Sigma_{nn}^{H}\left(k\right)=\frac{1}{2}\left[\Sigma_{nn}\left(k,E_{kn}\right)+\Sigma_{nn}^{*}\left(k,E_{kn}\right)\right]$$
for the diagonal elements. Note that Eq. (Equation 9) now contains the quasiparticle energies as energy arguments in accordance with the expression from perturbation theory.

Equation (Equation 2) (with $\Sigma \to \Sigma^{H}$) is then solved iteratively until self-consistency in the density n(r) is reached. During these iterations, the matrix $\Sigma^{H}$ is held fixed. Technically, we add and subtract $v_{\mathrm{xc}}\left(r\right)\psi_{kn}\left(r\right)$ in Equation (Equation 2) and fix the matrix $\Sigma^{H}-v_{\mathrm{xc}}$. This has the advantage that we have to intervene in the DFT code only once when we augment the Hamiltonian by $\Sigma^{H}-v_{\mathrm{xc}}$. Furthermore, the keeping of $v_{\mathrm{xc}}$ in the program run in the usual way enables using the same construction of the LAPW basis set (which relies on $v_{\mathrm{xc}}$) as in a standard DFT calculation.

This completes the first QSGW iteration. The resulting new eigensolutions $\varphi_{kn}\left(r\right)$ and $\epsilon_{kn}$ then serve as the mean-field starting point for the next iteration and so on. Therefore, in each QSGW iteration, there is an inner self-consistency cycle for the electron density. The outer QSGW self-consistency cycle iteratively optimizes the mean-field starting point defined by $\Sigma^{H}$. The outer loop has reached self-consistency when the energies $\epsilon_{kn}$ do not change anymore. One then runs a final GW calculation with the full non-Hermitian Σ in Equation (Equation 2).

## 3. Computational Details

We adopt the experimental crystal structures taken from the Inorganic Crystal Structure Database (ICSD) [24,25]. The lattice constants are listed in Table 1. Self-consistent DFT groundstate calculations are performed employing the all-electron full-potential linearized augmented-plane-wave (FLAPW) code fleur [26]. Exchange-correlation effects are treated within the local-density approximation (LDA) [27]. For the GW step (both one-shot and QSGW), we employ the spex code [9], which utilizes the same LAPW basis as fleur. The main computational parameters are presented in Table 2. The electronic self-energies are calculated using carefully tested Brillouin zone sampling k-point grids. The grid density is a very important parameter, in particular for metallic systems. To prevent core charge leakage, the states specified in the row LOs are treated explicitly as valence states by complementing the muffin-tin basis with auxiliary local orbitals. A total of 100 (occupied and unoccupied) states are involved in summations over states for computing the polarizability and the correlation self-energy. In the QSGW calculations, the self-energy Σ(k,ϵ) is calculated on the basis of the 40 lowest eigenstates, giving rise to a 40 × 40 self-energy matrix per k point and energy ϵ. On average, eight QSGW iterations are sufficient to reach converged values for all materials. Table 2 lists the k-point grids used for each system. The band structures presented in the next sections will contain the quasiparticle energies $E_{kn}$ obtained from Equation (Equation 2) for all grid k points that happen to lie on the high-symmetry k-point path. In addition, we show the k-resolved spectral functions Equation (Equation 6). To obtain smooth intensity plots not only along the energy axis but also along the k path, we employ Wannier interpolation for all elements of the self-energy matrix in Equation (Equation 6), which gives Σ(k,ϵ) everywhere in the Brillouin zone. The maximally localized Wannier orbitals are constructed with the Wannier90 library [28] involving *s* and *p* orbitals for Li, Be, Na, and Mg, and *s*, *p*, and *d* orbitals for K, Ca, Sr, Rb, and Cs.

## 4. Results and Discussion

Valence bandwidths as computed with different computational approaches, as well as available experimental estimates, are summarized in Table 3. Experimental bandwidths can be found for seven of the nine simple metals investigated here. A comparison shows that the DFT bandwidths of Li, Na, K, Ca, and Sr are strongly overestimated. The magnitude of the overestimation is ≈0.6 eV. We find an opposite case in Be, where the LDA bandwidth is, in fact, smaller than in the experiment. We refer here to the smaller of the two bandwidths shown in Table 3. Be crystallizes in a hexagonally close-packed (hcp) lattice, similarly to Mg bulk, which is the other exception. However, the experimental situation of Mg is unclear. There are two photoemission studies [29,30] that report very different bandwidth values. The older study reported bandwidths smaller than the LDA ones, in accordance with the common trend mentioned above. The newer study, on the other hand, yielded bandwidth values very close to the LDA prediction.

The overestimation of the bandwidths is usually attributed to the mean-field approximation of Kohn–Sham DFT, which does not allow for the formation of dressed quasiparticles. Dressed quasiparticles are expected to have a larger effective electron mass, which should decrease the curvature of the bands and, thus, the bandwidths. The GW approximation goes beyond the mean-field approximation and generally yields improved band structures with respect to Kohn–Sham DFT. As a first step, we have applied the most popular one-shot $G_{0}W_{0}$ approach, employing the LDA eigenstates as a starting point. In Table 3, we also list $G_{0}W_{0}$ values from Ref. [17]. Throughout, we find a reasonable agreement between the $G_{0}W_{0}$ bandwidths, with the ones from Ref. [17] being consistently larger.

The many-body renormalization by the GW self-energy leads to a significant reduction in the bandwidths by ≈0.3 eV for all materials except Be (and Mg), amounting to about 50% of the overestimation by LDA. Thus, the renormalization corrects the bandwidth in the right direction, although not sufficiently to reach experiment values. In the exceptional case of Be, LDA underestimates the bandwidth, and, indeed, the $G_{0}W_{0}$ renormalization correctly increases it.

The $G_{0}W_{0}$ self-energy correction is thus seen to improve the bandwidth in most cases, but a difference to the experimental values remains. It is then plausible that a repeated application of $G_{0}W_{0}$ may improve the results further. In fact, a self-consistent calculation within the QSGW method decreases the bandwidths of Li, Na, K, Ca, Rb, Sr, and Cs by ≈0.1 eV with respect to $G_{0}W_{0}$, further improving the agreement with available experimental values. Likewise, the QSGW calculation yields improved bandwidths for Be. The larger of the two bandwidths practically becomes identical to the experimental value; however, the smaller one ends up at quite a large distance of about 0.4 eV from the experiment. Our results are somewhat closer to the experiment than previous QSGW studies [20,38]. Reference [19] reports a QSGW bandwidth of 3.00 eV for Na, very close to ours. The bandwidths reported for Na (3.16 eV) and K (2.07 eV) in Ref. [20] deviate more strongly from the values calculated by us; however, it should be pointed out that the authors employed a variant of QSGW, which they called linearized QSGW (LQSGW) and which relies on a linearization of the GW self-energy around the Fermi energy. This variant is rather different from the traditional QSGW method we have used. Overall, our QSGW calculations yield theoretical bandwidths close to the experimental values.

The QSGW results indicate that the missing self-consistency is mainly responsible for the remaining overestimation of (most of) the bandwidths in the one-shot $G_{0}W_{0}$ approach. On the other hand, the GW self-energy also misses short-range correlation effects in the form of vertex corrections, which may also play a role in the description of simple metals. If this were the case, the common understanding of simple metals as representing approximate nearly-free electron gases with weak electronic correlation would have to be questioned. The issue has been investigated in Ref. [17] by an application of the eDMFT method to the simple metals. The eDMFT method includes short-range correlation effects in the impurity site described by the continuous-time quantum Monte Carlo method. However, since correlations are treated on a single impurity site, long-range correlations are not accounted for. The self-energy is local and thus misses the k dependence in contrast to the GW self-energy.

Table 3 lists the eDMFT bandwidths for comparison. They do represent an improvement over LDA and—to a lesser degree—over $G_{0}W_{0}$, but they are in many cases surprisingly close to QSGW, with a few exceptions: eDMFT underestimates the bandwidth of lithium by 0.4 eV, while QSGW approaches it to within 0.1 eV. The larger of the two Be bandwidths is predicted practically exactly by QSGW, while eDMFT shows a large underestimation of about 1.0 eV. The experimental bandwidth of K is about midway between the ones from QSGW and eDMFT.

Bulk Ca and Mg are interesting cases. For both systems, there are two experimental studies, one giving smaller and the other larger bandwidths. The former shows a better agreement with eDMFT, and the latter is closer to our QSGW results. Thus, for these systems, we cannot conclude which of the two approaches, eDMFT or QSGW, offers the better theoretical description.

In the cases where the eDMFT and QSGW results are different, we cannot uniquely attribute the difference to a specific property of any of the methods because the methods are different in many ways. Both methods are self-consistent, but only one (QSGW) captures the full k dependence of the self-energy, whereas the other (eDMFT) includes short-range correlations beyond GW. However, in most cases, the results of the two methods are rather similar. The improvement brought about by both methods with respect to one-shot $G_{0}W_{0}$ can, therefore, be attributed to the self-consistency included in both approaches. Hence, the present study confirms the common belief that the electrons in simple metals form nearly free electron gases with weak correlations.

In Figure 1, we present the electronic band structures for the simple metals. LDA bands are shown as solid black lines. The QSGW results are plotted in two ways. First, the red points mark the real parts of the (complex-valued) quasiparticle energies $E_{kn}$ (Equation (Equation 2)) at k points that are members of the k-point set specified in Table 2. Second, the QSGW spectral functions (Equation (Equation 6)) are plotted as intensity plots in blue. Wannier interpolation is used to obtain smooth spectral functions in k. It is remarkable how well Wannier interpolation works here, bearing in mind that the self-energy of Equation (Equation 3) is a non-local function. Its matrix elements with respect to the localized Wannier functions do not fall off exponentially with distance as in the case of the Kohn–Sham Hamiltonian. In fact, if we look closely, we can make out some slight spurious oscillatory behavior caused by the Wannier interpolation, for example, in the conduction bands of Rb around P-H-P.

The spectral functions exhibit a broadening of the quasiparticle bands, which stems from the finite lifetime of the quasiparticle excitations. The broadening is stronger for bands farther away from the Fermi energy. It vanishes at the Fermi energy because the scattering phase space vanishes in this limit. The explicit quasiparticle energies (red points) trace the quasiparticle bands and mostly fall in the middle of the broadened bands, although not always because the quasiparticle peaks can have an asymmetric line shape.

Apart from the bandwidth reduction in the bcc and fcc simple metals, the LDA and QSGW band structures on the whole are rather similar. Therefore, LDA yields a satisfactory description of the electronic structure in simple metals for most purposes. Only in Sr and Ca does the QSGW method have a stronger effect on the band dispersion around the Fermi energy, which should also affect the form of the Fermi surface to some extent.

## 5. Conclusions

We have investigated the electronic structure of the simple (alkali and alkaline earth) metals Li, Be, Na, Mg, K, Ca, Rb, Sr, and Cs with the one-shot $G_{0}W_{0}$ and the self-consistent QSGW method. In particular, we focused on a well-known problem of Kohn–Sham DFT (LDA), namely that the bandwidths of the simple metals are overestimated. Specifically, we observe an overestimation for the bcc and fcc crystals (Li, Na, K, Ca, and Sr—no experimental reference values available for Rb and Cs) but not for the hcp metals (Be and Mg).

The origin for the overestimation of the bandwidths (e.g., in the case of Li, 3.46 eV vs. 3.0 eV in the experiment) lies in the approximate mean-field treatment of exchange and correlation in Kohn–Sham DFT. In fact, if we apply a one-shot $G_{0}W_{0}$ correction to the self-consistent LDA solution, we find that the resulting quasiparticle bands correct the bandwidths in the right direction (e.g., 3.22 eV for Li); however, a difference from the experimental values remains.

The remaining difference may be due to the starting-point (LDA) dependence of $G_{0}W_{0}$. Another possible reason might be the missing vertex corrections in the GW approximation, i.e., its inadequacy in treating short-range (or strong) correlations. We have addressed the first issue by applying the QSGW method, which, being self-consistent, is free of the starting-point dependence. Indeed, our QSGW calculations bring the theoretical bandwidths even closer to the experimental reference values (e.g., 3.10 eV for Li). From the results, we conclude that the bandwidth overestimation of Kohn–Sham DFT is mainly due to long-range correlations, which are well described by the GW self-energy approximation. In other words, the simple metals can be regarded as nearly-free electron gases.

Of course, a small difference to the experiment still remains, which could be attributed to missing vertex corrections in GW. However, it should also be pointed out that whenever there are two experimental estimates (Be, Na, and Ca), the experimental values differ from each other as much as the theoretical QSGW values from the experiment.

The two hcp metals (Be and Mg) are exceptions. Kohn–Sham DFT does not exhibit an overestimation of the bandwidths. Interestingly, neither $G_{0}W_{0}$ nor QSGW have a strong effect on the band structure in these cases. To be more precise, there are two experiments for Mg giving rather different values of the bandwidth (we refer here to the largest bandwidth). The older measurement would indicate a bandwidth overestimation by LDA Kohn–Sham DFT, whereas the experimental bandwidth of the more recent measurement is very close to the LDA prediction. The QSGW bandwidth lies just in-between the two experimental estimates.

We also presented theoretical band structures calculated from LDA and QSGW. In addition to the quasiparticle bands, we plotted the k-resolved QSGW spectral functions, which not only show the quasiparticle band dispersions but also the lifetime band broadening. Wannier interpolation was used to create smooth spectral functions along k. We see that the lifetime broadening grows with the distance from the Fermi energy in both directions. At the Fermi energy, the lifetime broadening vanishes, and the quasiparticle peaks become delta peaks. Higher-order correlation effects, such as plasmon satellite features, are not observed in the considered energy interval.

## Figures and Tables

**Figure 1 nanomaterials-12-03660-f001:**
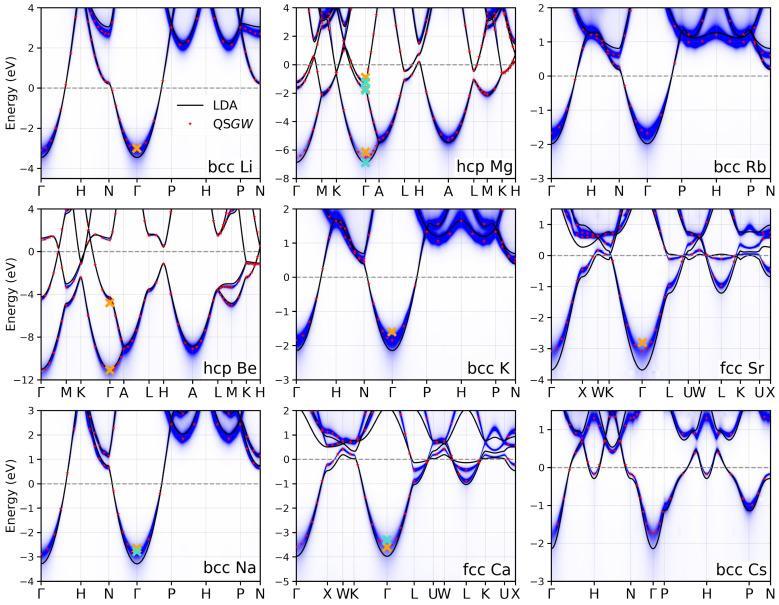
Electronic band structures of the simple metals calculated with LDA (solid black line) and QSGW (red stars). The Wannier-interpolated QSGW spectral functions are shown as intensity plots in blue. The band broadening scales with the inverse excitation lifetime and vanishes at the Fermi energy (0 eV). The orange crosses mark experimental binding energies. If two experimental values are available (see Table 3), the turquoise cross shows the more recent measurement.

**Table 1 nanomaterials-12-03660-t001:** Experimental crystal structure parameters taken from the inorganic crystal structure database (ICSD) [24,25].

	Li	Be	Na	Mg	K	Ca	Rb	Sr	Cs
ICSD-ID	44367	1425	196,972	76,748	44,670	44,348	44,869	76,162	42,662
type	bcc	hcp	bcc	hcp	bcc	fcc	bcc	fcc	bcc
a (Å)	3.5100	2.2858	4.2250	3.2093	5.3280	5.5884	5.6970	6.0849	6.4650
c (Å)		3.5843		5.2103					

**Table 2 nanomaterials-12-03660-t002:** Computational parameters.

	Li	Be	Na	Mg	K	Ca	Rb	Sr	Cs
$R_{\mathrm{MT}}$ (a${}_{0}$)	2.29	2.00	2.40	2.29	2.50	2.40	2.50	2.40	2.40
$G_{\max}$(a${}_{0}^{-1}$)	4.70	5.00	4.70	5.00	4.50	5.00	4.50	5.00	4.50
k grid	16 × 16 × 16	12 × 12 × 10	12 × 12 × 12	12 × 12 × 10	12 × 12 × 12	12 × 12 × 12	10 × 10 × 10	12 × 12 × 12	14 × 14 × 14
LOs			2s 2p	2p	3s 3p	3s 3p	4s 4p	4s 4p	5s 5p

**Table 3 nanomaterials-12-03660-t003:** Valence bandwidths (in eV) obtained from LDA, $G_{0}W_{0}$, QSGW, LQSGW, and eDMFT compared to experimental reference values.

	Li	Be		Na	Mg			K	Ca	Rb	Sr	Cs
LDA	3.46	11.02	4.26	3.29	6.87	1.62	1.28	2.16	3.98	1.99	3.68	2.14
LDA ${}^{1}$	3.46	11.03	4.28	3.30	6.89	1.65	1.31	2.15	3.98	1.99	3.70	2.15
$G_{0}W_{0}$	3.22	11.31	4.41	3.04	6.58	1.63	1.24	1.95	3.60	1.78	3.21	1.80
$G_{0}W_{0}$ ${}^{1}$	3.39	11.37	4.48	3.15	6.66	1.68	1.29	2.00	3.79	1.86	3.39	2.00
QSGW	3.10	11.14	4.36	2.90	6.52	1.64	1.21	1.88	3.60	1.68	3.10	1.74
QSGW ${}^{2}$				3.00								
LQSGW ${}^{3}$				3.16				2.07				
eDMFT ${}^{1}$	2.60	10.12	4.41	2.84	6.18	1.85	0.82	1.42	3.24	1.81	3.05	1.70
exp	3.0 ${}^{4}$	11.15	4.8 ${}^{6}$	2.65 ${}^{7}$	6.15	1.79	0.9 ${}^{9}$	-1.6 ${}^{11}$	-3.3 ${}^{12}$			
		11.15 ${}^{5}$		2.78 ${}^{8}$	6.89	1.79	1.23 ${}^{10}$		-3.6 ${}^{13}$		2.8 ${}^{13}$	

^1^ Ref. [17], ^2^ Ref. [19], ^3^ Ref. [20], ^4^ Ref. [5], ^5^ Ref. [31], ^6^ Ref. [32], ^7^ Ref. [33], ^8^ Ref. [34], ^9^ Ref. [29], ^10^ Ref. [30], ^11^ Ref. [35], ^12^ Ref. [36], ^13^ Ref. [37].

## Data Availability

All calculated data are contained within the article. The open source software packages fleur and spex are available for download at www.flapw.de, accessed on 3 October 2022, both distributed under the MIT license.

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
