# Peer review of "Quasiparticle Self-Consistent GW Study of Simple Metals"

_nanomaterials, 2022, doi:10.3390/nano12203660_

Round 1

Reviewer 1 Report

This paper reports quasiparticle self-consistent GW (QSGW) calculations for metallic systems, which are not commonplace, in response to some recent controversy. This area is not quite my specialty area so I found it a very informative and an interesting read.
It is very well presented and throughly describes the theory behind this and similar approaches, which I found helpful as a one-stop methodology shop.
Subsequent application to simple metals and discussion of results are well thought out and equally informative. 
I think this paper will be a valuable addition to the literature.

Author Response

We thank the Referee his/her positive evaluation of the manuscript.

Author Response

We thank the Referee for his/her positive evaluation of the manuscript. In the following, we give a point-by-point response to the Referee's comments.   --------------------------   - In the current manuscript, only the bandwidth is concerned. The conclusion could be more convincing if various observables are compared, cross-validating the conclusion. This is of course beyond the scope of this work. Could the authors at least add some comments and perspectives on that?      Reply: This would indeed be a worthwhile topic for another study, but it would go beyond the scope of the present work (as the Referee rightly says). In fact, many interesting quantities (the effective masses, the Fermi surface, the electronic state lifetimes, etc.) could be extracted from the bandstructure information. Unfortunately, the mentioned cross-validation of results would hardly be possible due to the rather scarce data available in literature (both experimental and theoretical). Our goal is to demonstrate that the GW method does serve as a suitable tool to study the electronic properties of simple metals and to interpret photoemission measurements. We see our study also as a motivation for experimental groups to reinvestigate the bandstructures of this seemingly simple family of materials due to its relevance for fundamental condensed matter theory and many-body physics. This may then also open up the possibility of cross validation.   --------------------------   - Could the authors also add a plot to make the results more intuitive? For example, to help myself read the data, I plotted the figure below, in which the vertical axis is the obtained bandwidth minus that obtained from LDA. From the figure, some features become more obvious. For example, for elements larger than Na, there seems to be an overall better agreement of eDMFT over both of the GW methods. How would we interpret these? Is it because the short-range correlation plays a much more significant role for heavy elements?   Reply: We thank the Referee for taking the effort of preparing the plot and for the suggestion. We have been thinking about it, but we are not so convinced about the idea because of the following reasons. Firstly, we think that the table gives the most “honest” rendition of the results, which is why such a comparison is usually done in tabulated form in literature. Secondly, a graphical representation will always give a bias to the interpretation, depending on how the plot is prepared, and might even be misleading. The diagram that the Referee has plotted might be taken as an example: One might see an oscillatory behaviour in the plotted curves, which is  deceptive, however. The curves give the difference of two theoretical bandwidths (that difference has no direct physical interpretation) and the x axis does not represent a continuous variable. Also note that there are two Be and three Mg bandwidth values! In this sense, the representation is a bit misleading. Furthermore, we would not go along with the Referee’s interpretation of the data that for systems larger than Na, eDMFT gives a better description. For many systems, there are two experimental bandwidths. While one set of experimental values might confirm the referee’s observation, the other set of experimental values actually shows a much better agreement with QSGW. Therefore, due to the experimental uncertainty, it must remain an open question which theoretical method gives a more realistic description. Again, we can hope that high-quality experimental bandwidths will become available in the future.   --------------------------

- If it is clear, could the authors explain why their G0W0 results are different from those in [17], as they have explained for the difference in QSGW results? I noticed that the LDA results are the same as those in [17].

Reply:
The Referee makes a very good point. In contrast to DFT (LDA), the GW calculations involve many more computational parameters and therefore are well known to be more sensitive to the computational setup. An analysis of the deviations would require a detailed knowledge of the used computational parameters and of the implementation used in Ref. 17. However, Ref. 17 does not give sufficient information to carry out such an analysis. Of course, we do not want to speculate. Therefore, we did our best to determine well-converged parameters based on extensive convergence tests and provide these computational parameters to make it possible for others to reproduce the results.
--------------------------   - Could the authors further clarify the renormalization by the GW self-energy or add references?   Reply: To address this request, we have added the following paragraph to the theory part, which is hopefully helpful for the reader   The GW self-energy, Eq. (3), constitutes the leading-order term of an expansion of the self-energy with respect to the screened Coulomb interaction potential [*]. Plugging the first term of Eq. (4) into Eq. (3) yields the exact non-local exchange potential of the Hartree-Fock method. The remainder of Eq. (4), when inserted into Eq. (3), gives rise to a non-local dynamical scattering potential, which describes a large part of the many-body correlation effects. It is also responsible for the emergence of lifetime effects and plasmon satellites.   [*] Martin, R.M.; Reining, L.; Ceperley, D.M.  Interacting Electrons: Theory and Computational Approaches; Cambridge University Press: Cambridge, 2016.   -------------------------   - Could the authors add some references for the “common belief” in line 60?   Reply: The Referee asks for a reference to the fact that simple metals are often treated or regarded as nearly free electron gases. Here, we point to an article by G. D. Mahan (the author of several monographs, e.g., “Many-particle physics”) published in the Encyclopaedia Britannica: https://www.britannica.com/science/crystal/Conductivity-of-metals In short, when forming a crystal, the valence electrons of the simple metal atoms become conduction electrons that can freely move in-between the ion lattice. These conduction electrons can be treated as a non-interacting electron gas within the Sommerfeld theory.  We may also cite Ref. [17]: "In particular, in this letter we show that the elemental metals with partially occupied s orbitals, which are usually assumed to be nearly-free-electron metals, are in fact moderately correlated, thus forcing a reconsideration of long-held notions about these simple metals.”

Reviewer 3 Report

The article titled 'Quasiparticle self-consistent GW study of simple metals ' studied a standard method to calculate the electronic band structure of simple metals (alkali and alkaline earth metals) from first principles. The simulation model is reasonable and the calculated results are appealing, mainly including the confirmation of a common belief that simple metals can be regarded as nearly free electron gases with weak electronic correlation. The authors give detailed verification and rich comparison, ending up with the clear conclusion that the applicability of the GW method in correction is well demonstrated. Hence I would recommend the publication of this manuscript.

Author Response

We thank the Referee for his/her positive evaluation of the manuscript.

Reviewer 4 Report

The author provided an extensive study of metals using the GW method and comparing it with G0W0. The scientific results were expected, however, this could be a relevant contribution to the literature.

I suggest the paper for publication after minor corrections to the figure. The light green cannot be easily distinguished respect to the normal green, please change color. Since the band structures have different k-path due to the different symmetries, it will be easier for the reader to add in the figure or in the caption the relative crystal structure of the elements as bcc, fcc and hcp.

Author Response

We thank the Referee for his/her positive evaluation of the manuscript and for his/her suggestions to improve the quality of the plots. In the revised manuscript, we have changed the bandstructure plots according to the Referee's recommendations: two colors (orange and turquoise) for the experimental data are now used instead of (light and dark) green — we hope that the crosses are now better visible — and the crystal structure type (bcc, fcc, hcp) is now included in the labels.